

# Modulation of glacier ablation by tephra coverage from Eyjafjallajökull and Grimsvötn volcanoes, Iceland: an automated field experiment

Rebecca Möller[1,2], Marco Möller[1,3], Peter A. Kukla[2], and Christoph Schneider[3]

[1]Department of Geography, RWTH Aachen University, Aachen, Germany
[2]Geological Institute, Energy and Minerals Resources Group, RWTH Aachen University, Aachen, Germany
[3]Geography Department, Humboldt-Universität zu Berlin, Germany

*Correspondence to:* Rebecca Möller (rebecca.moeller@geo.rwth-aachen.de)

**Abstract.** This Article reports results from a field experiment investigating the influence of volcanic tephra coverage on glacier ablation. These influences are known to be significantly different from those of moraine debris on glaciers due to the contrasting grain size distribution and thermal conductivity. Influences of tephra deposits on glacier ablation have hardly been studied so far. For the experiment, artificial plots of two different tephra types from Eyjafjallajökull and Grimsvötn volcanoes were installed on a snow-covered glacier surface of Vatnajökull ice cap, Iceland.Ablation was automatically monitored along with atmospheric variables and ablation on a non-tephra covered reference site over the summer season 2015. For each of the two volcanic tephra types, three plots ($\sim$1.5 mm, $\sim$8.5 mm and $\sim$80 mm) were monitored. After limiting the records to a period of reliable measurements, a 50-days dataset of hourly records was obtained, which can be downloaded from the Pangaea data repository (https://www.pangaea.de; doi:10.1594/PANGAEA.876656). The experiment shows a substantial increase of ablation under the $\sim$1.5 mm and $\sim$8.5 mm tephra plots when compared to uncovered conditions. Only under the thick tephra cover some insulating effects could be observed. This result is in contrast to other studies which depicted insulating effects for much thinner tephra coverson bare-ice glacier surfaces. Differences between the influences of the two different petrological types of tephra exist but are small.

## 1 Introduction

Deposits of sedimentary materials on the surface of glaciers are known to have significant influence on glacier melt as they alter the energy exchange processes at the surface (e.g., Nicholson and Benn, 2013; Mattson et al., 1993). The thickness of the layer controls whether the dominant influencing factor at the glacier surface is the decrease of albedo or the increase of thermal resistance (Möller et al., 2016). The former implies an increase of the energy gain to the glacier from solar radiation while the latter implies a decrease because of reduced heat conduction to the glacier surface. As a result, thin layers of supraglacial deposits lead to increased glacier melt, while thick layers imply decreased glacier melt or even insulation. With increasing layer thickness the increase of glacier melt peaks at the so-called effective thickness. Afterwards, glacier melt decreases continuously and returns to the level of uncovered conditions at the so-called critical thickness. Beyond this thickness, it decreases further





on and asymptotically approaches complete insulation (Adhikary et al., 1997).

A quantification of the influences from tephra coverage mostly relies on parametrisations. Often empirical in situ data are used for calibration. However, most of the so far developed formulations are designed to capture the effects of moraine debris deposits which are usually formed by layers with thicknesses in the order of meters or at least decimeters. Such studies dealing

with the relation between debris thickness and resulting modification of ablation are numerous in recent years (e.g., Collier et al., 2015; Juen et al., 2014; Pratap et al., 2015; Rounce et al., 2015). Volcanically active regions of the world in sub-polar and polar environments episodically experience the deposition of tephra on glacier surfaces after explosive volcanic eruptions. Volcanic tephra deposits show a wider range of depositional thicknesses, i.e. from sub-millimeter to meter scale, than moraine debris. They also feature distinctly different thermal properties (Brock et al., 2007). Studies dealing with the relation between

tephra thickness and the intensity of induced ablation change are remarkably less numerous than those dealing with moraine debris, even if supraglacial tephra deposits are known to significantly influence glacier surface processes and mass balance (e.g., Kirkbride and Dugmore, 2003; Möller et al., 2014; Nield et al., 2013). So far, only two recent studies carried out a systematic, quantitative investigation of the influence of tephra deposits of varying thickness on glacier ablation (Dragosics et al., 2016; Möller et al., 2016). However, both studies only rely on results obtained over very short periods. The experiments

covered periods of only 17 (Dragosics et al., 2016) or 13 days (Möller et al., 2016), respectively. Moreover, the experiment of Dragosics et al. (2016) was carried out in an ex situ, non-local environment under controlled, partly laboratory-like conditions. Here, we present a dataset of automated, continuous measurements of meteorological conditions and on-glacier snow ablation under artificially installed plots of volcanic tephra of different type and thickness. The measurements were obtained from a field experiment which was carried out on Vatnajökull ice cap, Iceland, over the summer season 2015. Snow melt rates under

different thicknesses of tephra during days with and without precipitation are compared to illustrate the variability of ablation with tephra thickness and the influence of different meteorological conditions.

## 2   Field experiment

### 2.1   Study site

The field experiment was carried out at an elevation of $\sim$970 m a.s.l. (above sea level) on Tungnaárjökull (64.3253°N, 18.0476°W), a glacier which is part of western Vatnajökull ice cap, Iceland (Fig. 1a). The site was situated on a slightly inclined surface, facing approximately west-southwest. It was characterized by wind-compacted snow coverage with a homogeneous depth of $\sim$2.7 ± 0.2 m around the site according to snow-depth probing. Layering of the snow pack was not well-pronounced and snow density showed little variability over the vertical profile with an integrated mean of $\sim$410 kg m$^{-3}$.





## 2.2 Design and setup

The field experiment was set up in order to quantify the influence of different types and ticknesses of volcanic tephra on snow ablation and to provide the opportunity to relate measured ablation to meteorological conditions. For this purpose a set of six different artificial plots of tephra coverage with a diameter of 0.7 m were installed at the study site. Three of these plots were made from tephra of Eyjafjallajökull volcano (EYV) and the other three from tephra of Grimsvötn volcano (GRV) (Fig. 1a). Both types of tephra were deployed at thicknesses of $\sim$1.5 mm, $\sim$8.5 mm and $\sim$80 mm. This was done by weighing out tephra material according to its bulk density (1,276 kg m$^{-3}$ for EYV and 791 kg m$^{-3}$ for GRV) as deployment by thickness is not feasible in the mm-scale. The three thicknesses were meant to approximately match the effective thickness (1.5 mm) and the critical thickness (8.5 mm) of the tephra and a thickness under which a dominance of insulation can be considered. The thickness values were chosen according to results of a short, 13-days field experiment by Möller et al. (2016) which was carried out on bare glacier ice using tephra of GRV.

Contiguous to the tephra plots, standard meteorological parameters were measured and recorded by an automatic weather station (AWS). Air temperature and relative humidity at two levels (initially 0.3 m and 1.1 m above snow, but increasing according to ablation), wind speed and direction, liquid precipitation and incoming and reflected shortwave radiation were measured along with ablation. In order to also facilitate automatic measurements on the tephra plots, an aluminium structure for sensor installation was mounted (Fig. 1b). Over each of the six plots ultrasonic height gauges for continuous monitoring of ablation were installed. Over the two $\sim$80 mm plots sensors for surface temperature measurements were installed in addition. Table 1 gives an overview of all sensor and measurement specifications for both the AWS and the tephra plots. The ablation measurement at the AWS provides a reference representing non-tephra covered conditions.

## 2.3 Tephra sampling

Tephra sampling for deployment of the six different plots was done directly at the calderas of EYV and GRV (Fig. 1a) in order to warrant the pristine character of the material. At EYV the tephra was acquired from inside the caldera (63.6314°N, 19.6373°W). This sampling was carried out on 7 May 2015. At GRV the tephra originates from rocky outcrops at the southern caldera rim (64.4061°N, 17.2741°W). Here, sampling was done on 8 May 2015. At both locations, the tephra was taken from active geothermal areas.

## 2.4 Measurements and data preparation

The experiment was brought into service on 10 May 2015 and delivered the first full-day record on 11 May 2015. From then on, the measurement records provided hourly means or samples from the sensors described in Table 1 until 8 September 2015, when ablation was so much advanced that the aluminium structures, holding the sensors above the tephra plots in place, collapsed on 9 September 2015. Collapsing was easily identifiable from unreasonable radiation and distance measurements. For



the actual purpose of the experiment, i.e. for studying the influences of tephra coverage on ablation, the records had to be narrowed down to a period without snow cover on top of the tephra. The selection of the suitable period is based on measured surface temperatures on the tephra packs of the two ∼80 mm plots (Fig. 2).

Surface temperature is generally closely related to the daily cycles of air temperature and shortwave radiation. However, despite this close relation, snow or ice surfaces cannot exceed 0 °C. This implies that daily surface temperatures which follow a regular above-zero cycle, indicate a completely snow or ice-free surface. In the case of our field experiment, this is the case for the period after 15 June 2015 (Fig. 2). Up until this date, sub-zero surface temperatures prevail despite the presence of daily air-temperature cycles, which regularly exceed 0 °C. This indicates at least partly snow covered conditions on the surfaces of the tephra plots.

From 4 August 2015 onwards, the daily cycles of surface temperature start to become irregular. In addition, the periodic, substantially positive offsets of surface temperature over air temperature, which occur consistently over the 15 June to 3 August period, were replaced by rather irregular, predominantly negative offsets (Fig. 2). This combination of observations suggests that the tephra packs start to disintegrate, giving room to snow or bare-ice outcrops which destroy the homogeneous surfaces of the tephra plots. Over a homogeneous, low-albedo tephra cover shortwave radiation adds considerably to the energy gain at the surface and thus drives surface temperatures far above the air-temperature level. Over a rather patchy tephra cover with high-albedo bare-ice outcrops the integrated energy gain due to absorbed shortwave radiation is much lower. In addition, the surface temperature of the outcrops is capped at 0 °C. The integrated surface temperature of the tephra plots might thus lie well below the air-temperature level.

Based on these considerations, we limit the field-experiment dataset with all its measurement records to the 50-days period 15 June to 3 August (Fig. 2). The final dataset contains hourly data for all meteorological parameters as measured at the AWS (Fig. 3a). Moreover, it contains hourly data from all seven ablation sensors, i.e. distance measurements over the six tephra plots and over the reference site at the AWS (Fig. 3b).

For illustration a comparative analysis of the ablation rates was carried out. To facilitate this analysis, running 24-hour differences, i.e. running daily ablation rates, were calculated for the data of each of the seven ablation sensors whenever distance measurements exist at all six tephra plots and at the reference site. The latter is considered in order to assure full comparability of the 24-hour ablation values. These running 24-hour differences are also part of the final dataset of the field experiment.

## 3  Results

Ablation measurements over the field-experiment period considered in the dataset (15 June to 3 August 2015) reveal a loss of 2.25 m of snow cover at the reference site and between 2.21 m and 2.97 m at the tephra plots (Fig. 3b). During the entire period the study site showed snow coverage. The high ablation sumsled to increased measurement uncertainty in the course of the study period due to a probable extension of the sensors' measurement areas beyond the borders of the individual tephra plots and erosion of the previously homogenous artificially set up tephra layer. Nevertheless, the running daily ablation rates, i.e. the



slopes of the ablation curves, show rather little variability with time, even if temporary increases are observable during end of June and during mid July. When comparing the different ablation curves to each other it appears that the ablation rates tend to harmonize over the second half of July, suggesting an incipient disintegration of the different tephra packs presumably due to erosion by meltwater.

Major disturbances are present in the ablation curves of two of the GRV tephra plots (∼8.5 mm and ∼80 mm) in mid July (Fig. 3b). On 14 July the measured distance at the ∼8.5 mm GRV tephra plot increased by ∼0.20 m, followed by an increase of ≥0.15 m at the ∼80 mm GRV tephra plot on 16 July. These disturbances coincide with a major rain event (Fig. 3a). It can thus not be ruled out that partial destructions of the tephra plots and of the upper layers of the snowpack occurred at this date and distorted all subsequent distance measurements at the six tephra plots.

The running daily ablation rates in relation to tephra thickness (Fig. 4) mostly confirms the findings of previous studies (Kirkbride and Dugmore, 2003; Mattson et al., 1993; Möller et al., 2016). At the thin (∼1.5 mm) tephra plots ablation was substantially increased by a factor of $1.49 \pm 0.88$ (mean ± one sigma over time) under EYV tephra and by a factor of $1.51 \pm 0.71$ under GRV tephra. At the tephra plots which were meant to match the critical thickness of the tephra (∼8.5 mm) ablation was equal to uncovered conditions under EYV tephra ($1.00 \pm 0.61$) and slightly increased under GRV tephra ($1.17 \pm 0.57$). However, at

the thick tephra plots (∼80 mm) the observed ablation did not match expectations. Under EYV tephra ablation was close to uncovered conditions ($0.98 \pm 0.73$) and under GRV tephra only a slight insulation effect was present ($0.85 \pm 0.59$). The rather high standard deviations, however, suggest a considerable, misleading influence of sporadic, anomalously high and potentially erroneous values. An assumption, which is supported by the distinctly more moderate medians of 0.93 (EYV) and 0.76 (GRV) (Fig. 4), which indicate insulating conditions under both ∼80 mm tephra covers. Nevertheless, the rather high ablation at the

two sites with ∼80 mm tephra cover suggests a substantially different behaviour of snowpacks under tephra coverage than of bare glacier ice under tephra coverage.

Distinct differences were observed between ablation rates during periods with and without precipitation (Fig. 4). On wet days the increase of ablation under the thin tephra covers is much more pronounced than on dry days. The substantial increase of ablation on wet days is in strong contrast to short-term measurements by Möller et al. (2016) on bare glacier ice where no

increase due to precipitation was found at all. Ablation under the ∼8.5 mm tephra covers is also higher on wet days than on dry days. This implies that the critical thickness of wet tephra is generally higher than that of dry tephra. The strength of the small insulation effect at the thick ∼80 mm tephra plots is, however, independent of the allocation to dry or to wet days.

Meteorological conditions during the field-experiment period (15 June to 3 August 2015) illustrate average summer conditions (Fig. 3a). Air temperature mostly fluctuates between $0\,^{\circ}\mathrm{C}$ and $+2\,^{\circ}\mathrm{C}$ with few outliers and shows a mean of $+2.1 \pm 1.4\,^{\circ}\mathrm{C}$

(mean ± one sigma). The mostly undisturbed daily shortwave radiation cycles suggest little cloud coverage. Accordingly, total rainfall over the period only sums up to 40.2 mm. However, high wind speeds of $5.65 \pm 3.34\,\mathrm{m\,s^{-1}}$ (mean ± one sigma) with peak-wind periods reaching $12–19\,\mathrm{m\,s^{-1}}$ might have led to considerable undercatch of precipitation by the tipping-bucket rain gauge (Sugiura et al., 2006).



## 4 Data description and availability

The final dataset is organized in one single *csv* file which is available for download (doi:10.1594/PANGAEA.876656) from the Pangaea earth and environmental sciences data repository (https://www.pangaea.de/). It contains 2,904 hourly samples of 18 variables. Among these, the six variables related to ablation measurements over the artificial tephra plots are limited to 1,200 hourly samples only (cf. section 2.4). The structure of the file is outlined in Table 2.

## 5 Summary and outlook

A field experiment, studying the influences of different types of volcanic tephra on snow ablation, was conducted on Vatnajökull ice cap, Iceland, in summer 2015. Two types of Icelandic tephra were compared, one from Eyjafjallajökull volcano and one from Grimsvötn volcano. Both tephras were sampled right before the start of the experiment at the calderas of the respective volcanoes. For the experiment, three different artificial plots of different thickness (∼1.5 mm, ∼8.5 mm and ∼80 mm) were installed from both tephras. Ablation at all six tephra plots and at an uncovered reference site was monitored automatically over the summer season as were surface temperature on the two ∼80 mm tephra plots and concurrent atmospheric variables. The experiment was run from mid May to mid October. During this time, it could fulfil the purpose it was meant for over a 50-days period and delivered a dataset of hourly resolution, for the period 15 June to 3 August. This final dataset comprises records of air temperature and relative humidity at two levels, wind speed and direction, rainfall, incoming and reflected short-wave radiation, ablation (in terms of distance from sensor to surface) over a non-tephra covered reference site and over the six tephra plots and surface temperature at the two ∼80 mm tephra plots. To illustrate the outcome and the successful completion of the field experiment, we presented a comparison of ablation rates under the different tephra plots.

Ablation showed substantial median increases at the two ∼1.5 mm tephra plots (∼17% under Eyjafjallajökull tephra and ∼40% under Grimsvötn tephra). However, ablation was also considerably increased at the ∼8.5 mm Grimsvötn tephra plot (median of ∼11%), which contrasts results of previous studies on bare-ice glacier surfaces. Insulation was small even under the thick ∼80 mm plots (median reductions of ∼7% under Eyjafjallajökull tephra and ∼24% under Grimsvötn tephra). This also stands in contrast to earlier bare-ice results, where almost full insulation was found under comparably thick tephra covers. The increase of ablation on days with rainfall under thinner tephra covers is markedly higher than on days without rainfall. This is in contrast to bare-ice conditions, where no ablation-increase is present on rainfall days at all. This finding leaves room for further investigations. Overall, the quantitative influence of Grimsvötn tephra on ablation was found to be stronger than that of Eyjafjallajökull tephra.

In conclusion, the experiment delivers a dataset which clearly illustrates that the influences of a supraglacial tephra cover on glacier ablation are considerably different, depending on the surface of the glacier, i.e. snow or bare ice. To the knowledge of the authors, this dataset is the first of its kind, as no continuous, automated ablation measurements over different types and thicknesses of volcanic tephra on snow surfaces have been conducted so far. Together with the recorded, contemporaneous meteorological conditions, this unique ablation dataset allows further in-depth studies of the influences of weather conditions



on sub-tephra snow melt. Moreover, it can readily be included as calibration or validation dataset in broader studies on the influences of supraglacial particle covers on ablation.

*Acknowledgements.* The field experiment was funded by grants no. SCHN680/6-1 and KU1476/5-1 of the German Research Foundation
5 (DFG). We thank the Vatnajökull National Park administration for granting permission to carry out the experiment and the associated tephra sampling at Grimsvötn caldera.





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





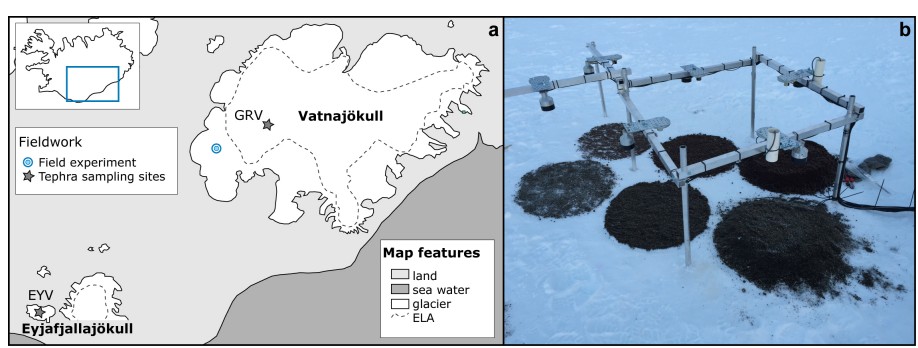

**Figure 1.** Overview of the field experiment. The locations of tephra sampling at the calderas of Eyjafjallajökull volcano (EYV) and Grimsvötn volcano (GRV) and the location of the field experiment are shown in (a). The installation of the field experiment itself is shown in (b). The three plots in the foreground are covered by EYV tephra and the three plots in the back by GRV tephra.





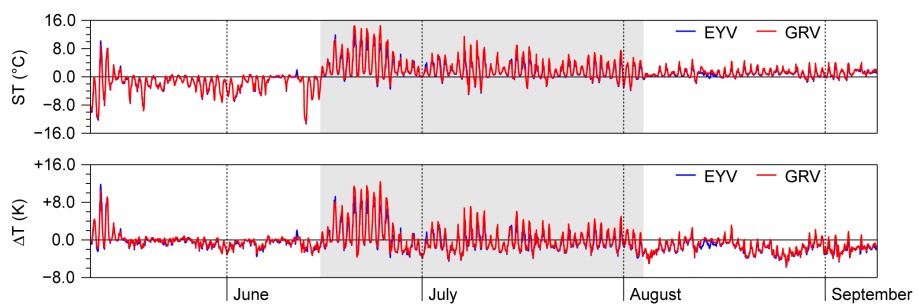

**Figure 2.** Records of measured hourly surface temperatures at the two ∼80 mm tephra plots (upper graph) and calculated differences between these surface temperatures and air temperatures measured at the automatic weather station (lower graph) over the period 11 May to 8 September. The air temperatures are calculated as the mean of upper and lower air temperature sensor at the AWS. The types of tephra (EYV for Eyjafjallajökull volcano and GRV for Grimsvötn volcano) on which the surface temperatures were measured are indicated by greyscale. The grey shading in the center of the time series indicates the period considered in the final dataset, i.e. 15 June to 3 August.



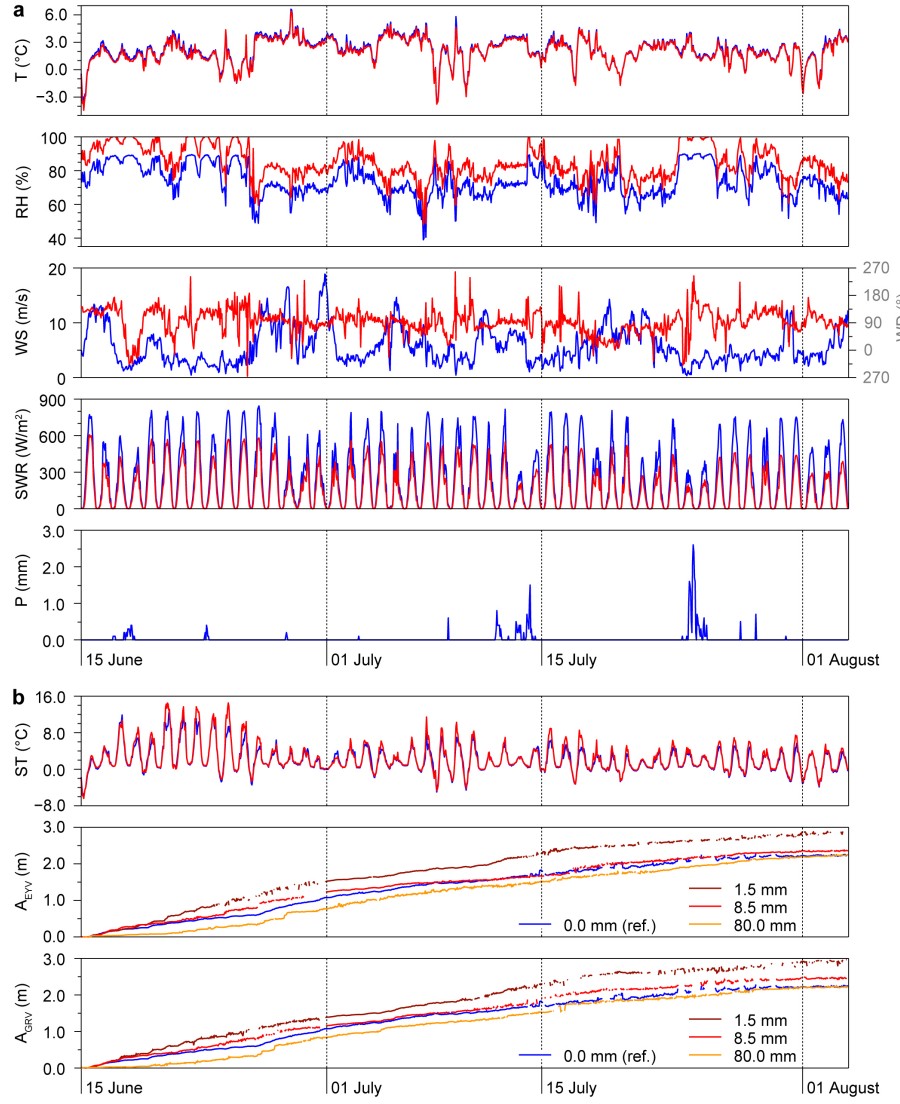

**Figure 3.** Hourly records of the measurements of all sensors installed at the automatic weather station are shown in (a) and measurements of all sensors mounted at the field experiment installation are shown in (b). Records are shown for the period 15 June to 3 August. For air temperature (T) and relative humidity (RH) the records of the upper sensor (blue line) are shown together with those of the lower sensor (red line). Wind speed (blue line) is shown together with wind direction (red line); note the different y-axes here. Incoming shortwave radiation (SWR, blue line) is shown together with reflected shortwave radiation (red line). As precipitation (P) only the liquid fraction has been measured. Surface temperatures (ST) are shown for the ~80 mm plots of tephra from Eyjafjallajökull volcano (EYV, blue line) and from Grimsvötn volcano (GRV, red line). Cumulative ablation (A) is shown over the different plots (indicated by colour codes) of EYV tephra and GRV tephra.



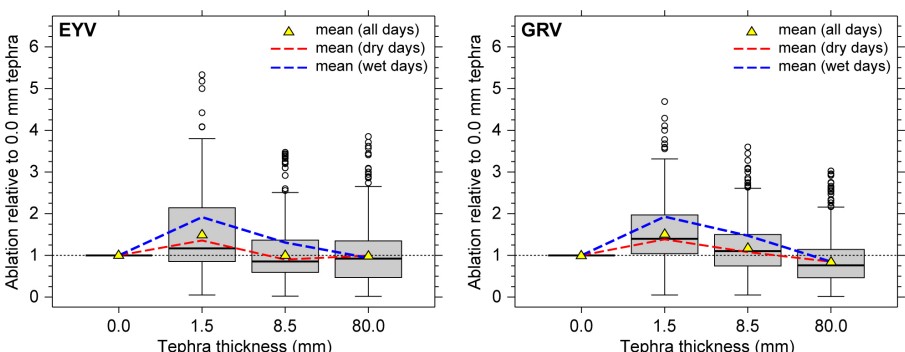

**Figure 4.** Running 24-hour ablation rates at the different plots of tephra from Eyjafjallajökull volcano (EYV) and from Grimsvötn volcano (GRV) relative to ablation rates measured at the non-tephra covered reference site. The box plots give an overview of the data spread across all running 24-hour values recorded during the field-experiment period (15 June to 3 August). Outliers are indicated as open circle symbols. The mean values of relative ablation changes over the field experiment period are indicated by yellow triangles, the mean values over wet (precipitation >0.1 mm) and dry (precipitation ≤0.1 mm) days are shown as colour-coded line graphs.



**Table 1.** Measured quantities at the field experiment installation and at the automatic weather station. For each variable the type of the sensor is given along with its uncertainty and the type of data aggregation over each one hour record interval.

| Variable | Sensor | Uncertainty | Aggregation |
|---|---|---|---|
| Air temperature | Vaisala HMP35C | $\pm 0.4\,K$ | Average |
| Relative humidity | Vaisala HMP35C | $\pm 3\%$ | Average |
| Incoming SW radiation | Campbell Scientific CS300 | $\pm 5\%$ | Average |
| Reflected SW radiation | Campbell Scientific SP1110 | $\pm 5\%$ | Average |
| Rainfall | RM Young 52203 | $\pm 2\%$ | Total |
| Wind speed | RM Young 05103 | $\pm 0.3\,m\,s^{-1}$ | Average |
| Wind direction | RM Young 05103 | n.a. | Sample |
| Ablation (reference) | Campbell Scientific SR50 | $\pm 1\,cm$ | Sample |
| Surface temperature | Campbell Scientific IRTS-P | $\pm 0.3\,K$ | Average |
| Ablation (tephra plots) | Campbell Scientific SR50A | $\pm 1\,cm$ | Sample |





**Table 2.** Structure of the overall *.csv* dataset showing records obtained during 11 May 2015 to 08 September 2015. Records belonging to the field-experiment period (15 June 2015 to 03 August 2015) are flagged accordingly (*1* instead of *0* in column 2). Missing records or ablation records over the artificial tephra plots outside the field-experiment period are indicted by *-9999*. The short names of the measured variables are shown as given in the first header line of the *.csv* file. The associated units are shown together with the text given in the second header line of the *.csv* file (in parentheses). Ablation is always shown in form of cumulative values. Short names indicate whether the records were measured at the automatic weather station (AWS), over tephra from Eyjafjallajökull volcano (EYV) or over tephra from Grimsvötn volcano (GRV).

| Column No | Variable | Short name | Unit |
|:---:|:---:|:---:|:---:|
| 1 | Date and time | TIMESTAMP | YYYY/MM/DD HH:MM (DateTime) |
| 2 | Identifier for field-experiment period | FLAG | n.a. (Y1-N0) |
| 3 | Air temperature at upper sensor | AWS_AT_upp | $^\circ$C (DegC) |
| 4 | Relative humidity at upper sensor | AWS_RH_upp | % (%) |
| 5 | Air temperature at lower sensor | AWS_AT_low | $^\circ$C (DegC) |
| 6 | Relative humidity at lower sensor | AWS_RH_low | % (%) |
| 7 | Wind speed | AWS_WS | m/s (m/s) |
| 8 | Wind direction | AWS_WD | $^\circ$ (Deg) |
| 9 | Rainfall | AWS_R | mm (mm) |
| 10 | Incoming SW radiation | AWS_SWR_in | $W/m^2$ ($W/m^2$) |
| 11 | Reflected SW radiation | AWS_SWR_refl | $W/m^2$ ($W/m^2$) |
| 12 | Ablation at bare glacier | AWS_ABL | m (m) |
| 13 | Surface temperature on 80.0 mm EYV tephra plot | EYV_ST_800 | $^\circ$C (DegC) |
| 14 | Ablation at 80.0 mm EYV tephra plot | EYV_ABL_800 | m (m) |
| 15 | Ablation at 8.5 mm EYV tephra plot | EYV_ABL_85 | m (m) |
| 16 | Ablation at 1.5 mm EYV tephra plot | EYV_ABL_15 | m (m) |
| 17 | Surface temperature on 80.0 mm GRV tephra plot | GRV_ST_800 | $^\circ$C (DegC) |
| 18 | Ablation at 80.0 mm GRV tephra plot | GRV_ABL_800 | m (m) |
| 19 | Ablation at 8.5 mm GRV tephra plot | GRV_ABL_85 | m (m) |
| 20 | Ablation at 1.5 mm GRV tephra plot | GRV_ABL_15 | m (m) |