# Peer review of "Modulation of glacier ablation by tephra coverage from Eyjafjallajökull and Grimsvötn volcanoes, Iceland: an automated field experiment"

_Earth System Science Data, 2017_

## Referee Comment (RC1) · J. Lenaerts (Referee) · 2 Aug 2017

The authors present an observational dataset collected on Vatnojökull ice cap, Iceland. The dataset consists of meteorological and glacier ablation observations under various tephra coverage depths and types. This dataset is useful to calibrate mass and energy balance models to account for tephra coverage, and to understand the impact of tephra thickness on glacier meteorology and ablation. The data are very nicely presented and I don't have any major objections against publication. I only have some minor comments that the authors can consider before publication.

P1, L7: ...three plots of variable thickness ($\sim$1.5 mm....

[Figure]

P1, L13: can you quantify 'small'?

P2, L2: ...on parameterisations, where in situ data are...

P3, L2: this might need some more details. How large were the plots? Also, what is the possible impact on the relatively small size of the plots on the measurement errors? There will be always be a certain level of contimination of the true tephra signal because the radiation instruments also see the surrounding snow. Would that be somehow quantifiable?

P4, L31: sums led

P5: perhaps a brief section on the impact on meteorology, mainly the surface energy balance, would be very worthwhile. For instance, what is change in net shortwave radiation? What are the temperature differences? You can simply refer to Figure 3, but mentioning some numbers would be good.

Figure 1b: perhaps good to increase brightness of the picture.

---

## Referee Comment (RC2) · C. Mayer (Referee) · 25 Aug 2017

This manuscript presents an interesting data set on snow pack evolution underneath different defined tephra coverages. This dataset has the potential for interesting investigations in relation to engergy transfer on tephra covered glacier surfaces and might form the basis for further interesting studies. However, there is one major issue, which I want to raise and which needs to be used for a detailed revision of the manuscript: the measurements carried out on the test plots are not ablation measurements, but distance measurements. In contrast to ice surfaces, where the assumption of constant density holds rather well, snow packs will change considerably during the ablation pe-

riod. If there exists a snow pack of 2.7m with a mean density of 410 kg/m$^3$ the elevation change of the surface is by no means directly transferable to ablation rates. There are a number of other processes involved (compaction, refreezing, rain percolation, melt, lag in run-off) which makes it complicated to convert surface elevation change into ablation rate. Therefore, these observations cannot be directly related to ablation conditions of sub-tephra ice surfaces. It needs a careful revision of the manuscript to present these complicated processes in the correct framework. The fact that water will not be present at the surface of the snow layer, together with the soft texture of the snow probably enables rather stable conditions for the experiment. The test plots would have probably been destroyed much earlier on an ice surface. It is a pity that no camera was installed in addition to the SR50 sensors. This could have provided valuable additional information about the conditions of the test plots and might have explained some of the irregularities found in the data set. Also, thermistors in the snow pack would have provided necessary information about percolation conditions and possibly compaction. This might be a useful recommendation for future experimental setups. Our detailed experiments (Juen et al., 2013) demonstrated that the surface morphology of spatially restricted tephra layers can change very fast, preventing the collection of sensible data. I might be useful to refer to this publication during revision.

Minor comments:

Introduction: Reference to Östrem, 1959 is missing. Many of the basic findings were already mentioned in this first publication on sub-debris ablation and credit should be given to him.

P. 2, L. 2: "influences from tephra cover" on what?

P.2, L.4: This is not correct. Most parameterisations for debris cover are also valid for debris thicknesses in the order to centimetres. Evatt et al., 2016 even provides a closed formulation for the gradual evolution of the debris cover from dust to m-scale.

P.2, L.8: see comment above.

P.2, L.13/14: This is incorrect. Juen et al., 2013 provides a very detailed study about energy transfer through debris cover, including the effect for tephra layers. Even though the observation period is very short, the findings are of interest for your study.

P2, L. 29: The vertical density profile would be interesting to see, in order to evaluate the ablation conditions.

P.3, L13/14: At what initial height above the surface were wind and radiation sensors installed. This important for the calculation of the turbulent fluxes.

P.3, L.18: If you use the SR50 sensor without additional information it is only a distance measurement, not an ablation measurement. In the case of underlying ice, the relation with ablation is rather straightforward, but for snow this is rather complicated.

P.3, L.32: The information about the total elevation change at the site and the depth of the holes for the aluminium structure would be useful for understanding the situation.

P.4, L.6: I doubt that this is correct. It rather depends on the specification of the temperature sensor. Does it provide the mean temperature (IR radiation) across the footprint, or does it report maximum and/or minimum temperatures? Also, it depends on the distance between the sensor and the surface, because the footprint might be larger than the sample plots for long distances (i.e. late in the season). There needs to be a more detailed description.

P.4, L. 24ff: As discussed above, the distance measurements are not identical to ablation measurements: Therefore, the presentation needs to be altered.

P.4, L.30: The maximum elevation change of 2.97m is larger than the snow thickness in the region of the experiment. This means that at least parts of the test sites were snow free at the end of the data set period. Are there any indications for the loss of the snow cover in the data?

P.5, L.19-21: This is a crucial finding and requires more attention. Because the distance measurements are not directly related to ablation, there might be other processes, which influence the elevation change underneath the thick tephra covers. One possible reason could be that the test plots were too small. Snow is a media with an open pore space, in contrast to ice. Warm temperatures, wind and high humidity can considerably change the internal structure in the snow pack and a surface cover of 70 cm in diameter definitely does not inhibit lateral influence through the open pore space.

P.5, L.22-25: Again, this is related to the porosity of the snow pack. Rain cannot penetrate the ice surface and results in run-off with very little effect on ice melt. For a snowpack, rain percolates into the snow layers and might cause compaction, melt, run-off or refreezing, in dependence on the air temperature, amount and temperature of the water and the temperature of the snow pack.

P.5, L.25: This has nothing to do with wet or dry tephra, but with the amount of rainfall and the according temperatures. Our experience shows that under melting conditions the thin tephra layers are always wet, due to their porosity and ability to absorb water.

Fig. 3: It would be useful to present also the daily distance measurements, not only the cumulated. This data would provide a better understanding of the daily variations. Already in Fig. 3 it is evident that the thick tephra layers reduce the surface lowering considerably during the initial phase with intact test plots (until mid-July).

References: Evatt, G.W., Abrahams, D., Heil, M., Mayer, C., Kingslake, J., Mitchell, S.L., Fowler, A.C. & Clark, C.D., 2015. Glacial melt under a porous debris layer, J. Glaciol., 61 (229), 825-836.

Juen, M., Mayer, C., Lambrecht, A., Wirbel, A. & Kueppers, U., 2013. Thermal properties of a supraglacial debris layer with respect to lithology and grain size, Geografiska Analer, Series A, Physical Geography, doi:10.1111/geoa.12011.

Östrem, G., 1959. Ice melting under a thin layer of moraine, and the existence of ice cores in moraine ridges. Geografiska Annaler, 41, 228–230.

---

## Author Comment (AC1) · 15 Oct 2017

PREAMBLE:

We very much thank the two reviewers for their thorough analysis of our article and for their valuable comments, annotations and suggested improvements. They had been carefully considered and most of them are accounted for in the revised version of the manuscript. Also, we widely follow comments and corrections regarding the writing of the manuscript. Answers and explanations to all detailed questions and annotations raised by the reviewers are provided in the following. Apart from the suggested changes we deleted Table 2 from the manuscript. This was done in order to avoid doubling of information which is already given in a much more detailed way in the metadata on the PANGAEA website. Moreover, we now manage to not exceed the recommended number (five) of figures and or tables in an ESSD publication.

ANSWERS TO COMMENTS BY REVIEWER 1:

Minor comments:

RC1-1: P1, L7: ...three plots of variable thickness (_1.5 mm....

AC: Changed accordingly.

RC1-2: P1, L13: can you quantify 'small'?

AC: A real quantification of "small" in very few words is not feasible. Hence, we rephrased the respective sentence with respect to the reviewer's comment to: "Differences between the influences of the two different petrological types of tephra exist but are negligible compared to the effect of tephra coverage in principal."

RC1-3: P2, L2: ...on parameterisations, where in situ data are...

AC: Changed to "...on parameterisations, that use in situ data for calibration."

RC1-4: P3, L2: this might need some more details. How large were the plots? Also, whatis the possible impact on the relatively small size of the plots on the measurementerrors? There will be always be a certain level of contimination of the true tephrasignal because the radiation instruments also see the surrounding snow. Would thatbe somehow quantifiable?

AC: The size of the plots (diameter of 0.7 m) is already reported in the manuscript. However, we now also included a discussion about the potential influences of the rather small plot size on the quality of the measurements. This mostly happened in response to the major comment of reviewer 2. The respective text passage can be found in the new paragraph on P5 L30ff in the revised version of the manuscript.
RC1-5: P4, L31: sums led

AC: Corrected accordingly.

RC1-6: P5: perhaps a brief section on the impact on meteorology, mainly the surface energybalance, would be very worthwhile. For instance, what is change in net short-waveradiation? What are the temperature differences? You can simply refer to Figure 3, butmentioning some numbers would be good.

AC: The paragraph describing the climate conditions during the field-experiment period has been extended according to the reviewer's suggestion. It now also includes key numbers of air temperature differences between the two sensors, albedo and net short-wave radiation. Information on the most frequently occurring wind direction has also been added. The respective text passages read: "Thereby, mean ($\pm$ one sigma) air temperature gradients between lower and upper sensor amount to +0.20$\pm$0.15 K m 1. Daily albedo means decrease from $\sim$0.71 during the first week of the field-experiment period to $\sim$0.58 during its last week. The associated daily mean of net shortwave radiation fluxes is 86.0$\pm$22.4 W m 2." and "The by far most frequently occurring wind directions (ENE to ESE) resemble the katabatic flow direction down the western slope of Vatnajökull."

RC1-7: Figure 1b: perhaps good to increase brightness of the picture.

AC: We suppose that this comment is mostly meant with respect to the surface structure of the tephra plots. We therefore brightened the picture and increased contrast.

ANSWERS TO COMMENTS BY REVIEWER 2:

Major comment:

RC2-1: There is one major issue, which I want to raise and which needs to be used for a detailed revision of the manuscript: the measurements carried out on the test plots are not ablation measurements, but distance measurements. In contrast to ice surfaces, where the assumption of constant density holds rather well, snow packs will

change considerably during the ablation period. If there exists a snow pack of 2.7m with a mean density of 410 kg/m$^3$the elevation change of the surface is by no means directly transferable to ablation rates. There are a number of other processes involved (compaction, refreezing, rain percolation, melt, lag in run-off) which makes it complicated to convert surface elevation change into ablation rate. Therefore, these observations cannot be directly related to ablation conditions of sub-tephra ice surfaces. It needs a careful revision of the manuscript to present these complicated processes in the correct framework. The fact that water will not be present at the surface of the snow layer, together with the soft texture of the snow probably enables rather stable conditions for the experiment. The test plots would have probably been destroyed much earlier on an ice surface. It is a pity that no camera was installed in addition to the SR50 sensors. This could have provided valuable additional information about the conditions of the test plots and might have explained some of the irregularities found in the data set. Also, thermistors in the snow pack would have provided necessary information about percolation conditions and possibly compaction. This might be a useful recommendation for future experimental setups. Our detailed experiments (Juen et al., 2013) demonstrated that the surface morphology of spatially restricted tephra layers can change very fast, preventing the collection of sensible data. I might be useful to refer to this publication during revision.

AC: The reviewer is fully right with his critical comments about the direct transferability of the snow depth measurements to ablation values. We accordingly revised the entire manuscript. The terminus "snow depth reduction" is now used where appropriate. We nevertheless keep the title as it is in order to report the aim of the field experiment to the reader at the first possible instance. In depth explanation about the missing direct comparability of snow depth reduction and ablation is now included in the introduction. Additional, explanatory details are now also given in various appropriate parts of the manuscript, i.e. in especially at the end of the introduction (P2 L26ff in the revised version of the manuscript: "Our measurements cannot be set directly equal to snow ablation, as a continuous monitoring of snow density beneath the tephra plots

simultaneous to the measurements of snow depth reduction were not carried out due to logistical limitations.") and in the results section (P5 L33ff in the revised version of the manuscript: "One conceivable explanation is the fact that pure snow ablation is masked by additional processes in the snow depth measurements. Snow depth reductions resulting from general settling and compaction of the snowpack as well as from metamorphism on the snow-crystal level definitely also imprint on the snow depth measurements. Moreover, the rather small horizontal extent of the tephra plots probably permits lateral influences of weather conditions on the snowpack beneath the plots."). However, we refrain from more in-depth discussions of the topic as ESSD is explicitly meant to document published datasets and not to further analyse or discuss those data. Regarding cameras for monitoring of the experiment it has to be noted that a camera system has been installed at the field experiment site, but unfortunately it stopped working due to a technical failure a few days after installation. Hence, there was the plan to monitor the site in order to get better insights into reasons for potential irregularities in the data, but it didn't work out. Related additional information has been added to the end of the Design and setup subsection (P3 L24ff in the revised version of the manuscript). Recommendations of the reviewer regarding improvements of the experimental setup are accounted for in an additional paragraph in the Summary and Outlook section (P7 L31ff in the revised version of the manuscript: "For potential future experiments, the results and our experiences in the field suggest that frequent snow profile analyses or at least snow density measurements over the experiment period would help to transfer the obtained snow depth measurements to snow ablation. However, this is logistically challenging, as is the suggestion of larger tephra plot diameters, which would better prevent the snow depth reduction measurements being influenced by lateral energy fluxes from the surface to the sub-tephra snowpack."). We thank the reviewer for the recommendation of the publication by Juen et al. (2013), which is now also referred to in the manuscript.

Minor comments:

RC2-2: Introduction: Reference to Östrem, 1959 is missing. Many of the basic findings were already mentioned in this first publication on sub-debris ablation and credit should be given to him.

AC: Reference to Østrem (1959) has been added according to the reviewer's suggestion.

RC2-3: P. 2, L. 2: "influences from tephra cover" on what?

AC: Explanatory information ("...on glacier melt...") has been added.

RC2-4: P.2, L.4: This is not correct. Most parameterisations for debris cover are also valid for debris thicknesses in the order to centimetres. Evatt et al., 2016 even provides a closed formulation for the gradual evolution of the debris cover from dust to m-scale.

AC: In the respective paragraph we now refer to "...thicknesses in the order of meters or at least deci- or centimeters." Also, additional explanatory reference is given to the model formulation of Evatt et al. (2016).

RC2-5: P.2, L.8: see comment above.

AC: See answer above.

RC2-6: P.2, L.13/14: This is incorrect. Juen et al., 2013 provides a very detailed study about energy transfer through debris cover, including the effect for tephra layers. Even though the observation period is very short, the findings are of interest for your study.

AC: The reviewer is right and additional reference is given to the important study of Juen et al. (2013). (see AC to RC2-1).

RC2-7: P2, L. 29: The vertical density profile would be interesting to see, in order to evaluatethe ablation conditions.

AC: As already stated in the text, the layering observed in the snow pack was not at all pronounced and hence no detailed snow profile observations have been carried out.

The given mean snow density was obtained from measuring the entire snow column. Apart from that, the permitted number of figures/tables in ESSD papers is too limited to include any additional figure.

RC2-8: P.3, L13/14: At what initial height above the surface were wind and radiation sensors installed. This important for the calculation of the turbulent fluxes.

AC: The initial height of the sensor (2.1 m above surface) has been added.

RC2-9: P.3, L.18: If you use the SR50 sensor without additional information it is only a distance measurement, not an ablation measurement. In the case of underlying ice, the relation with ablation is rather straightforward, but for snow this is rather complicated.

AC: The respective text passage has been changed accordingly (see also AC to RC2-1); it now reads: "The snow depth change measurement at the AWS provides a reference representing non-tephra covered conditions."

RC2-10: P.3, L.32: The information about the total elevation change at the site and the depth of the holes for the aluminium structure would be useful for understanding the situation.

AC: Here, we would like to refrain from fully following the reviewer's suggestion. Information about elevation (snow depth?) changes at the experiment site could easily be inferred from Figure 3b and thus no additional information should be necessary. Moreover, information about hole depths would just tell the reader that the structure collapsed with the lowermost parts of the aluminium tubes still inside the ice. We decided that this information can rather be conveyed by adding a short explanatory sentence. This has now been done at the respective text passage: "Collapsing happened with the lowermost parts of the structure still inside the ice (probably due to the mass centre lying too high above the ground)."

RC2-11: P.4, L.6: I doubt that this is correct. It rather depends on the specification of the temperature sensor. Does it provide the mean temperature (IR radiation) across the

footprint, or does it report maximum and/or minimum temperatures? Also, it depends on the distance between the sensor and the surface, because the footprint might be larger than the sample plots for long distances (i.e. late in the season). There needs to be a more detailed description.

AC: The temperature sensor provides mean temperatures across the footprint. However, the employed sensor type features an especially narrow view field, so that disturbances by snow-covered surroundings are minimized.We therefore think, that regular above-zero cycles of surface temperature over the course of a day really indicate fully exposed, snow-free tephra plots. We slightly rephrased the respective text passage in order to make things clearer here: "Surface temperature is generally closely related to the intra-day cycles of air temperature and shortwave radiation. However, despite this close relation, snow or ice surfaces cannot exceed 0°C. This implies that surface temperatures which follow a regular above-zero intra-day cycle, indicate a completely snow or ice-free surface."

RC2-12: P.4, L. 24ff: As discussed above, the distance measurements are not identical to ablation measurements: Therefore, the presentation needs to be altered.

AC: Done as suggested (see also AC to RC2-1).

RC2-13: P.4, L.30: The maximum elevation change of 2.97m is larger than the snow thickness in the region of the experiment. This means that at least parts of the test sites were snow free at the end of the data set period. Are there any indications for the loss of the snow cover in the data?

AC: The loss of the snow cover can only be traced from the albedo of the uncovered glacier surface. This can be calculated from incoming shortwave radiation and reflected shortwave radiation over the uncovered glacier surface. Both variables are part of the presented dataset. Albedo values indicate complete removal of the snow pack during the second week of August. However, it is reasonable to assume that the snowpack under some of the tephra plots was already removed before this date. A note on this

has been added to the respective text passage: "During almost the entire period the study site showed snow coverage. Only for the plots with ~1.5mm tephra coverage it is reasonable to assume that the snowpack beneath the plots got lost close before the end of the study period. For the reference site, complete snowpack removal occurred during the second week of August according to the measured albedo values."

RC2-14: P.5, L.19-21: This is a crucial finding and requires more attention. Because the distance measurements are not directly related to ablation, there might be other processes, which influence the elevation change underneath the thick tephra covers. One possible reason could be that the test plots were too small. Snow is a media with an open pore space, in contrast to ice. Warm temperatures, wind and high humidity can considerably change the internal structure in the snow pack and a surface cover of 70 cm in diameter definitely does not inhibit lateral influence through the open pore space.

AC: The reviewer is fully right here. We thankfully follow his suggestion and included a new paragraph dealing with this issue (P5 L32ff in the revised version of the manuscript: "This unexpected and thus important finding cannot be explained in full detail here because of limitations in the experimental setup. One obvious explanation is the fact that pure snow ablation is masked by additional processes in the snow depth measurements. Snow depth reductions resulting from general settling and compaction of the snowpack as well as from metamorphism on the snow-crystal level definitely also imprint on the snow depth measurements. Moreover, the rather small horizontal extent of the tephra plots probably permits lateral influences of weather conditions on the snowpack beneath the plots. Explanations beyond these influences cannot be given, because the pure, energy-balance controlled ablation signal cannot be isolated from measured snow depth reduction."). Moreover, we added suggestions for future experiments, here (P6 L4f in the revised version of the manuscript: " It is thus recommended that future experiment setups at least account for snow density variations in one or the other way.") and also in the Summery and outlook section (P7 L31ff in the

revised version of the manuscript: " For potential future experiments, the results and our experiences in the field suggest that frequent snow profile analyses or at least snow density measurements over the experiment period would help to transfer the obtained snow depth measurements to snow ablation. However, this is logistically challenging, as is the suggestion of larger tephra plot diameters, which would better prevent the snow depth reduction measurements being influenced by lateral energy fluxes from the surface to the sub-tephra snowpack."). In addition we wanted to justify the rather small size of the tephra plots by the fact that larger plots would unfortunately have been logistically impossible. For a potential transformation of all six plots of our field experiment to diameters of 200 cm, ~320 kg of tephra would have to be used in total. This is ~205 kg more than for our experiment with 0.7 m plots. A note on the logistically challenging transport issue is now also included in the Summary and outlook section (P8 L1f in the revised version of the manuscript: " Installing the six tephra plots with a diameter of 2.0m instead of 0.7m would have required the transport of over 320 kg of tephra (instead of ~115 kg) from the two sampling sites to the field experiment site.").

RC2-15: P.5, L.22-25: Again, this is related to the porosity of the snow pack. Rain cannot penetrate the ice surface and results in run-off with very little effect on ice melt. For a snowpack, rain percolates into the snow layers and might cause compaction, melt, run-off or refreezing, in dependence on the air temperature, amount and temperature of the water and the temperature of the snow pack.

AC: When it comes to differences between surface lowering of a snow pack under tephra deposits and surface lowering of glacier ice under tephra deposits, this explanation is certainly right. However, the respective paragraph of the text refers to relative changes of surface lowering under tephra deposits in comparison to uncovered conditions. And these are not explainable by any porosity issues, as both the snow pack under the tephra and the uncovered snow pack are equally porous. Nevertheless, we are thankful for the reviewer's comment as it reveals the ambiguousness of the respective text passage. We revised it in order to make it more clear and avoid further

misunderstandings. It now reads out as: "Distinct differences were observed between snow depth reduction rates during periods with and without precipitation (Fig. 4). On wet days the increase of snow depth reduction rates under the thin tephra covers compared to uncovered conditions is even more pronounced than it is on dry days. This finding is in clear contrast to short-term measurements by Möller et al. (2016) on bare glacier ice. Their study shows that on wet days sub-tephra ice ablation rates are even decreased when compared to uncovered conditions. The increase of snow depth reduction under the ~8.5 mm tephra covers compared to uncovered conditions is also higher on wet days than on dry days."

RC2-16: P.5, L.25: This has nothing to do with wet or dry tephra, but with the amount of rainfall and the according temperatures. Our experience shows that under melting conditions the thin tephra layers are always wet, due to their porosity and ability to absorb water.

AC: It could certainly be the case that the reviewer is right with his explanation. However, only observations are described here and no potential explanations should be given, as any analysis or discussion should be kept to a minimum in ESSD publications. Hence, we just described that the relative increase of snow depth reduction under the 8.5 mm tephra layer in comparison to uncovered conditions is higher on wet days than on dry days. However, we slightly rephrased the respective text passage to make these facts clearer to the reader (see AC to RC2-15).

RC2-17: Fig. 3: It would be useful to present also the daily distance measurements, not only the cumulated. This data would provide a better understanding of the daily variations. Already in Fig. 3 it is evident that the thick tephra layers reduce the surface lowering considerably during the initial phase with intact test plots (until mid-July).

AC: We would like to refrain from introducing more figure materials into the paper as we already reached the permitted amount of five figures and/or tables. Moreover, ESSD publications are explicitly meant to document published datasets, while detailed analyses or discussion of the data should not be included.